# Establishment of a *Coilia nasus* Spermatogonial Stem Cell Line Capable of Spermatogenesis In Vitro

**DOI:** 10.3390/biology12091175

**Published:** 2023-08-28

**Authors:** Kaiyan Gu, Ya Zhang, Ying Zhong, Yuting Kan, Muhammad Jawad, Lang Gui, Mingchun Ren, Gangchun Xu, Dong Liu, Mingyou Li

**Affiliations:** 1Key Laboratory of Integrated Rice-Fish Farming, Ministry of Agriculture and Rural Affairs, Shanghai Ocean University, Shanghai 201306, China; m210100083@st.shou.edu.cn (K.G.); zhangya@shou.edu.cn (Y.Z.); zhongy10@haid.com.cn (Y.Z.); m200100064@st.shou.edu.cn (Y.K.); jawadktk1293@gmail.com (M.J.); lgui@shou.edu.cn (L.G.); 2Key Laboratory of Exploration and Utilization of Aquatic Genetic Resources, Ministry of Education, Shanghai Ocean University, Shanghai 201306, China; 3Key Laboratory of Microecological Resources and Utilization in Breeding Industry, Ministry of Agriculture and Rural Affairs, Guangzhou 511400, China; 4Key Laboratory of Freshwater Fisheries and Germplasm Resources Utilization, Ministry of Agriculture, Freshwater Fisheries Research Center, Chinese Academy of Fishery Sciences, Wuxi 214081, China; renmc@ffrc.cn (M.R.); xugc@ffrc.cn (G.X.)

**Keywords:** *Coilia nasus*, spermatogonial stem cell line, spermatids, in vitro spermatogenesis, cryopreservation

## Abstract

**Simple Summary:**

Induction of sperm cell differentiation in vitro is a key technology for conserving germplasm resources. *Coilia nasus* is an anadromous fish with commercial value found in the Yangtze River in China. Overfishing and deteriorating ecological conditions have almost caused the extinction of the *C. nasus*’s natural resources in the past decade. To preserve the declining population of *C. nasus*, a stable spermatogonial stem cell line (CnSSC) of *C. nasus* was obtained. The cell line remained proliferative and maintained stable cell morphology, a normal diploid karyotype, and normal gene expression patterns for over a year. Additionally, the cells could differentiate into sperm. Our research results contribute to further research on endangered fish germplasm resources of significant value.

**Abstract:**

The process by which spermatogonial stem cells (SSCs) continuously go through mitosis, meiosis, and differentiation to produce gametes that transmit genetic information is known as spermatogenesis. Recapitulation of spermatogenesis in vitro is hindered by the challenge of collecting spermatogonial stem cells under long-term in vitro culture conditions. *Coilia nasus* is a commercially valuable anadromous migrant fish found in the Yangtze River in China. In the past few decades, exploitation and a deteriorating ecological environment have nearly caused the extinction of *C. nasus*’s natural resources. In the present study, we established a stable spermatogonial stem cell line (CnSSC) from the gonadal tissue of the endangered species *C. nasus*. The cell line continued to proliferate and maintain stable cell morphology, a normal diploid karyotype, and gene expression patterns after more than one year of cell culture (>80 passages). Additionally, CnSSC cells could successfully differentiate into sperm cells through a coculture system. Therefore, the establishment of endangered species spermatogonial stem cell lines is a model for studying spermatogenesis in vitro and a feasible way to preserve germplasm resources.

## 1. Introduction

Fish species are the most abundant of all vertebrates, but several are in danger of extinction due to severe population decreases [1]. As a consequence, the conservation study of fish germplasm resources is becoming more and more crucial to the preservation of fish in danger. Long-term preservation of fish germplasm plays an important role in the conservation of endangered fish [2,3] and the establishment of cell lines contributes to preserving genetic resources [4].

Spermatogenesis is the process by which mature sperm are continuously produced in the testis [5]. The process of spermatogenesis relies on spermatogonial stem cells (SSCs) [6]. The only adult stem cells in males capable of passing on genetic information to the next generation are SSCs [7,8], which can self-renew to maintain a sufficient amount and differentiate to mature functional sperm [9]. SSCs, however, make up a tiny portion of all testis cells [7,10]. Therefore, the exploration of the molecular mechanisms promoting SSC self-renewal and differentiation and the utilization of germinal stem cells has been greatly facilitated by using long-term culture methods for SSCs [11].

Previous studies showed that sperm cells have been induced in vitro from vertebrate stem cells [11]. In mice (*Mus musculus*), a 3D culture of pluripotent stem cells was used to replicate the entire process of spermatogenesis. After induction in vitro, functioning sperm and offspring were produced [12]. In Koi (*Cyprinus carpio haematopterus*), fibroblasts were reprogrammed into pluripotent stem cells by chemical reprogramming techniques using small-molecule drugs, and pluripotent stem cells underwent meiosis to produce sperm-like cells in vitro after differentiating into cells that resembled germ cells [13]. In medaka (*Oryzias latipes*), the spermatogonial stem cell line SG3 was cocultured in suspension with embryonic rainbow-colored trout gonadal (RTG) stromal cells to produce motile sperm [5]. Our research group also successfully established the long-term SSC line ObSSC from Chinese hook snout carp (*Opsariichthys bidens*) through in vitro culture, which can be cultured to differentiate into sperm [14]. The induced differentiation of fish germinal stem cells is important for exploring the potential mechanisms regulating spermatogenesis in fish and may also lay the foundation for fish germ cell transplantation.

*C. nasus* is an important fish species in China’s Yangtze River because of its nutritional value and delicate taste [15]. In recent years, *C. nasus* has nearly become extinct due to overfishing and changes in aquatic ecology [16]. Very little research progress has been made on its artificial breeding [17]. Up to now, the gonadal somatic cell line CnGSC has been isolated from *C. nasus* [1]. In the current research, we successfully established the *C. nasus* spermatogonial cell line (CnSSC) and investigated the in vitro sperm production capability of CnSSC with gonadal somatic cells induction. As a result, the establishment of spermatogonial stem cell lines from endangered species provides a foundation for further research on *C. nasus* reproduction and breeding as well as a donor for researching spermatogenesis in vitro.

## 2. Materials and Methods

### 2.1. Fish, Primary Cell Culture, and Subculture

Six-month-old *C. nasus* measuring 12 cm were raised at the Freshwater Fisheries Research Center of the Chinese Academy of Fishery Sciences. Samples of testicular tissues were extracted after disinfection and decapitation. After the extraction, testis tissue was washed 3 times and minced with sharp scissors in phosphate-buffered solution (PBS, PH 7.4) containing 1% antibiotics (streptomycin, 1000 µg mL^−1^; penicillin, 1000 IU mL^−1^). Following dissociation, the fragments were incubated for 1 h in 1 mL of trypsin-EDTA (28 °C; Gibco, Grand Island, NE, USA), according to a previous description [18]. Next, 1.5 mL of ESM2 medium was added to the cell separation solution to stop digestion, and then the cell suspension was centrifuged at 800× *g* for 10 min (28 °C) to remove the supernatant. The cells were resuspended by adding 1.5 mL of ESM2 medium. The suspension of prior cells was filtered through a 300-mesh sieve to obtain separated single cells and then transferred to 6-well plates covered with gelatin for adherent culture (28 °C). Cells that had been dissociated were cultured in ESM2 medium and the fat cells were carefully aspirated from the upper layer of the medium, changing new fresh medium daily for the first week. Larger embryoid bodies formed several days later were digested with trypsin-EDTA for 5 s and resuspended by adding new medium to distribute the cells in the plates. Following 15 passages of culture, stable CnSSC cell lines were obtained, and the culture medium was changed to EMS4 for subculture performed as described [19]. For ESM2, each 500 mL of DMEM (HEPES) (pH 7.8; Gibco, New York, New York, NY, USA) contains the following components: MEE (1.25 mL; 400 medaka embryos/mL; 0.5 mL for EMS4), bFGF (50 µL; 100 ng/mL; 10 µL for EMS4), FBS (75 mL; Gibco, New South Wales, NSW, Australia), sodium selenite (0.5 mL; stock 2 µM; Sigma, Burlington, MA, USA), fish serum (5 mL; Seabass; serum extraction process as described in [5]), 2-mercaptoethanol (2 mL; stock 50 mM; Sigma), pen/strep (all 100×; 5 mL; Gibco, New York, New York, NY, USA), L-glutamine, sodium pyruvate, nonessential amino acids. The culture medium was stored (4 °C) for up to 6 months after being filtered through filters (0.22 µm). These studies were performed in compliance with the Helsinki Declaration, and the SHOU Animal Care and Use Committee approved these studies under the authorization number SHOU-2023-031.

### 2.2. Chromosome Analysis

Chromosome analysis was performed according to the previously described steps [20]. CnSSCs were collected at passages 30, 50, and 70. Briefly, cells were cultured at 28 °C for 24 h after being seeded in 6-well dishes. Next, cells were exposed to the colchicine solution (0.1 μg mL^−1^) at 28 °C for 4 h and washed twice with PBS. The cells were collected with PBS, treated with hypotonic solution (75 mM KCl) for 0.5 h, washed twice with PBS, fixed with freshly prepared fixative (3:1 methanol/acetic acid), and mounted on clean glass slides. After air drying, stained slides with 5% Giemsa solution (PH 6.8, 28 °C) for 30 min and then washed slides with water. An oil immersion microscope (100× magnification) was used to observe and photograph the slides.

### 2.3. Gene Expression Analysis

RNA obtained from the cells using Trizol reagent (Sigma, Burlington, MA, USA) was reverse transcribed according to the protocol of the cDNA synthesis kit (TaKaRa, Kusatsu, Japan). RT-PCR was carried out using *C. nasus* gene-specific primers (Table 1), gonadal germ cells (*dazl*, *vasa*), stem cells (*nanog*), somatic cells *(clu*, *hsd3β*), and meiotic cells (*dmc1*, *rec8*). PCR was run as follows: 95 °C for 20 s, followed by 28 (β-actin) and 38 (rest) cycles of 95 °C for 30 s and 60 °C for 30 s, with an ending extension of 72 °C for 1 min based on the previous description [21]. The PCR products of the 20 μL reaction system were resolved on 1% or 2% of agarose gels with a DNA marker of 1 kb ladder.

### 2.4. Cryopreservation, Thawing, and Alkaline Phosphatase (AP) Staining of CnSSC

For cryopreservation, CnSSC cells at 90% confluence were collected in 2 mL cryovials. CnSSCs were digested using 0.25% trypsin-EDTA followed by a centrifugation of 3 min at 1200× *g*. Since vitrification is a method extensively used for cryopreservation, CnSSCs were resuspended in 1.5 mL of the storage protection solution that contained 10% dimethyl sulfoxide (DMSO, Sigma, D2650-100ML), 70% EMS4, and 20% FBS (Gibco, New South Wales, NSW, Australia). The cryovials were placed at −80 °C for gradient cooling (1 °C/min, 3 h), and then moved to liquid nitrogen, as described previously [4]. Immediately place the cryovials in a bath of 37 °C water for 2 min to thaw the cells, and after centrifugation at 800× *g* for 10 min, the cells (suspended in completely fresh EMS4) were seeded on a 24-well plate covered with gelatin. Alkaline phosphatase (AP) staining was used to test whether the cultured cells had stem cell activity as described [22]. Cells were seeded on a 24-well plate coated with gelatin and fixed in 4% paraformaldehyde for 10 min after CnSSCs had grown to 60% confluence, then washed 3 times with PBS. The cells were incubated in freshly prepared BCIP-NBT solution (Invitrogen, Waltham, MA, USA, N6547) for 12 h (28 °C) and washed 3 times with PBS. Glycerol (200 µL) was added to cover the cells, and the cells on the 24-well plate were observed under the microscope (20× and 40× magnification).

### 2.5. Immunofluorescence Staining of CnSSC

The round climbing slices were soaked in 75% alcohol for 12 h and then placed on the 24-well plate carefully with forceps after the alcohol evaporated. Cover slices with 0.1% gelatin for 2 h and the gelatin was aspirated and dried at room temperature for 6 h. CnSSCs were seeded on a 24-well crawler coated with gelatin and fixed in 4% paraformaldehyde for 10 min after CnSSCs had grown to 70% confluence, then washed 3 times with PBS (500 μL). Each well was permeabilized with 0.1% Triton-X 100 (200 µL, Solarbio, Beijing, China) for 10 min and washed 3 times with PBS (500 µL). Cells in each well were blocked in 5% BSA (200 µL, Sigma, V900933) for 30 min and washed once with PBST (500 µL). A quantity of 200 uL proliferating cell nuclear antigen (PCNA) antibody (mouse antibody, Sigma, P8825) or Vasa antibody (rabbit antibody, Invitrogen, Waltham, MA, USA, PA5-30749) was added to each well and treated for 2 h under room temperature (1:200 dilution in 5% BSA), as described previously [14]. Cells were cleaned 3 times in PBST (500 µL), and 200 uL TRITC secondary antibody of goat anti-mouse (Invitrogen, A11126) or FITC secondary antibody of goat anti-rabbit (Invitrogen, A11034) was incubated for 2 h in the corresponding cell wells, protected from light (1:200 dilution in 3% BSA). PBST washed cells 3 times, and 200 µL DAPI (Sigma, D9542) was added to each well for 15 min (1:300 dilution in PBST). After washing 3 times with PBST, the climbing slices were removed and dried, and then sealed on slides with the gold antifade reagent (1 µL, Invitrogen, Carlsbad, CA, USA).

### 2.6. Cell Transfection

CnSSCs were inoculated into 24-well plates covered with 0.1% gelatin and transfected with 2 µg pCVpr DNA after the cells had grown to 70% confluence using the GeneJuice reagent (Novagen, Darmstadt, Germany), as previously described [23]. The CnSSCs stably expressing RFP were obtained, and the construction procedure was basically the same as that of the ObSSCs stably expressing RFP, as mentioned previously [14].

### 2.7. Coculture of CnGSCs and CnSSCs

After harvesting by trypsinization, 10^4^ CnGSCs and 10^4^ RFP-positive CnSSCs were seeded into 24-well plates (covered with 0.1% gelatin) and cocultured at 28 °C with 500 μL of the EMS4 medium changed every 2–3 days without subculture. During a week of CnGSCs induction, the exfoliated cells were transferred to 6 cm plates to be suspended in culture with the medium changed daily. Cell morphological changes were observed continuously for 7 days under a Nikon ECLIPSE Ti inverted microscope (Tokyo, Japan) with a camera (10×, 20× and 40× magnification).

## 3. Results

### 3.1. Establishment of a Spermatogonial Stem Cell Line

The process of establishing a *C. nasus* SSC line (CnSSC) is shown in Figure 1. All gonadal tissues were dissociated using trypsin and seeded on the ESM2 plates (covered with 0.1% gelatin). Fat and cell masses were observed in the freshly dissociated suspension of *C. nasus* gonad cell suspension (Figure 2A). After 2 days of differential adherent culture, some of the cells attached to the plate, and the cell nucleolus was large (Figure 2B). Within a week, large numbers of cells attached to the plates and proliferated (approximately 90% confluency) (Figure 2C). With the passage of time, the formation of embryoid bodies (EBs) occurred, and the quantity and rate of cell proliferation increased (Figure 2D). The cell morphology of the cells dissociated around embryoid bodies showed a small size and an oval or polygonal shape (Figure 2E). Following a 216-day culture, these dissociated cells at passage 78 maintained an oval or polygonal shape, sparse cytoplasm and uniform morphology, which are typical morphologies of spermatogonia (Figure 2F). At passage 38, the majority (90%) of seeded cells showed positive staining for alkaline phosphatase (AP) (Figure 3A–C). After cryopreservation for 1 month, CnSSCs at passage 38 were reinitiated from the liquid nitrogen (Figure 3D). After thawing, CnSSCs were seeded on a 24-well plate coated with gelatin for staining. Every cell (passage 38) displayed alkaline phosphatase activity (Figure 3E,F).

### 3.2. Chromosome Analysis and Characterization of Spermatogonial Stem Cell Properties

The majority of CnSSC cells showed a diploid karyotype and had 48 chromosomes (Figure 4A). Apparently, CnSSC shows genetic stability during long-term culture. CnSSCs, spermatogonial stem cells, were identified by using transcripts of the germ cell markers *dazl*, *vasa*, the stem cell multipotent marker *nanog*, and the somatic cell markers *clu*, *hsd3β*. Experiments showed that CnSSCs transcribe *dazl*, *vasa*, and *nanog* (Figure 4B,C), while the Sertoli cell marker gene *clu* and Leydig cell marker gene *hsd3β* were not expressed, as was observed at the mRNA level (Figure 4D).

Immunofluorescence staining performed on the round climbing slices showed the expression of PCNA and Vasa in the undifferentiated CnSSCs (Figure 4F,I). Positive expression of PCNA indicates high proliferative activity (Figure 4H), and the germ cell marker Vasa was expressed in all cells of CnSSCs, thereby determining their germ cell origin (Figure 4K).

### 3.3. Sperm Production of CnSSCs In Vitro

At passage 38, the plasmid pCVpr was transfected into the CnSSCs to clearly identify the CnSSCs. The red fluorescent protein (RFP)-expressing CnSSCs were clonally expanded for over one month (Figure 5A–C), maintaining high AP activity (Figure 5D). RFP-expressing CnSSCs formed 3D-EBs in coculture with CnGSCs on the third day (Figure 5E–G). In order to mimic meiosis in vitro, the CnSSCs were cocultured with gonadal somatic cells of *C. nasus* to observe morphological changes in the cells. On the first day of culture, EMS4 medium containing fish serum, FBS, bFGF, and medaka embryo extract (MEM) was used to culture cells. We obtained spherical spermatids on day 5 of coculture that were morphologically similar to those derived from the testis (Figure 6A). On the 7th day of coculture, it was observed that the cells extended their tails to form sperm cells (Figure 6B). On day 12 of coculture, postmeiotic products were produced with long and thin tails by the CnSSC cell (Figure 6C).

To ascertain whether CnSSC cells proceed through meiosis, we evaluated the meiotic gene transcripts present in CnSSC cells through RT-PCR. Under normal culture conditions, the levels of mRNA for both genes were low in CnSSC cells (Figure 7). Notably, increased expression levels of both genes were found in CnSSC cells through coculture (Figure 7).

## 4. Discussion

In this study, we provide the first description of a normal spermatogonial stem cell line cultured from the endangered fish *C. nasus*. From five fish, the *C. nasus* spermatogonial stem cell line that was already grown for more than 80 passages over the span of a year in culture without apparent senescence was obtained and termed CnSSC. This cell line could be stably cultured for a long time by primary culture and produced sperm-like cells when cocultured with the *C. nasus* gonadal somatic cell line. The germline of *C. nasus* was identified using AP staining, karyotype analysis, gene expression patterns, and cellular immunohistochemistry. In medaka and orange-spotted grouper (*Epinephelus coioides*), alkaline phosphatase (AP) activity is a typical marker that has been applied to effectively identify pluripotency in ES-like cell cultures from stem cell lines [22,24,25]. While AP-positive cells showed an ES-like cell shape (small, low in cytoplasm, round, or polygonal), AP-negative cells presented a differentiated phenotype (extended, cytoplasm-rich) [22]. The cultured primary cell line has strong alkaline phosphatase activity, and 48 chromosomes were detected in the cell line, indicating a diploid karyotype consistent with the previously reported chromosome karyotype of male *C. nasus* [26]. CnSSCs, spermatogonial stem cells, were identified by using transcripts of the germ cell markers *dazl*, *vasa*, the stem cell multipotent marker *nanog*, and the somatic cell markers *clu*, *hsd3β*. *Dazl* has been characterized as a marker gene in *C. nasus* germ cells [21]. In addition, *vasa* is the first molecular marker of germ cells identified in fish and is abundantly expressed in germ cells of Nile tilapia, medaka, and rainbow trout [27,28,29]. *Nanog* is highly expressed in undifferentiated spermatogonial stem cells of medaka and zebrafish [30,31] and is important for maintaining spermatogonial stem cells in farmed carp [32]. Stem cells of the ovary from Chinese soft-shell turtles express the germ cell markers *dazl* and *vasa* and the stem cell multipotent marker *nanog* during in vitro culture [33]. The germ cell marker genes *dazl* and *vasa* and the stem cell marker gene *nanog* were abundantly expressed in CnSSCs, while the somatic cell marker genes *clu* [34] and *hsd3β* [35] were not expressed, which indicates that the cells are germinal stem cells. The expression of proliferating cell nuclear antigen (PCNA) and Vasa proteins in cells was verified through a cellular immunohistochemistry test at the protein level [36,37]. PCNA and Vasa proteins are strongly expressed in the established cell line, indicating that the cell line has the phenotype of C. nasus SSCs. After one month of cryopreservation, the CnSSCs display alkaline phosphatase activity and exhibit stable growth after 80 passages in culture for more than one year under defined culture conditions. The *C. nasus* spermatogonial stem cell line (CnSSC) was effectively established in vitro on the basis of evidence indicating that CnSSCs consistently exhibited SSC-like properties.

Although spermatogonia are capable of proliferating continuously, a suitable culture system is important for long-term cultivation. Stem cell culture systems in fish are complex and vary for different fish species [38], and the addition of different components to the culture medium is essential for spermatogonial stem cells to maintain stable transmission and cell stemness [5]. Spermatogonial stem cells exist in a special microenvironment of the gonads, where growth factors are released that are necessary for the proliferation of spermatogonial stem cells [11], and stable proliferation of SSCs can be maintained with a certain concentration of bFGF [39,40]. In Chinese hook snout carp (*Opsariichthys bidens*), a spermatogonial stem cell line that remained undifferentiated stably for a long period of time was obtained using no feeder layer but with the addition of trophic factors such as FBS, bFGF, 2-mercaptoethanol, embryo extract (MEE), and fish serum [14]. In this experiment, we mimicked the culture conditions of *Opsariichthys bidens* spermatogonial stem cells, and the CnSSCs still maintained a good proliferation status, providing a favorable condition for the subsequent research on spermatogenesis in vitro.

The process by which SSCs continuously go through mitosis, meiosis, and differentiation to produce gametes that transmit genetic information is known as spermatogenesis. Meiosis results in round spermatids, while sperm is formed during spermiogenesis. Previous studies have shown that spermatogenesis from SSCs in vitro is influenced by their microenvironment [41,42]. In medaka *(O. latipes)*, meiotic differentiation occurred and sperm were formed after long-term cultured SG3 cells were induced through coculture of the CnGSCs [5]. In the current study, meiotic gene expression analysis revealed that CnSSCs retained their meiosis ability after more than one year of cell culture. Space arrangement and 3D structuring of SSCs in the seminiferous tubules may be an option to support spermatogenesis [43]. In Chinese gudgeon (*Bostrychus sinensis*), premeiotic spermatogonia were isolated, which could differentiate into viable sperm through induction cultivation methods in vitro [44]. Interestingly, a coculture system creating a simulation of the testicular environment in vivo inducted CnSSCs into sperm-like cells. DNA meiotic recombinase 1 (Dmc1), encoded by *dmc1*, is the recombinase that performs meiotic recombination [45]. *Rec8,* found on the centromere and adjacent chromosome arms in prophase meiosis, has a conserved role in the initiation program of meiosis [46,47]. The meiosis cell marker gene *dmc1* and *rec8* were abundantly expressed, indicating that the CnSSC cell line maintains the ability to go through meiosis. Despite the successful induction of this cell line into sperm-like cells by coculture, the ability to recapitulate spermatogenesis in vitro requires further research, and further data are required to identify how biological processes differ and are similar in vivo and in vitro. In zebrafish, reproductive stem cell transplantation increased donor sperm production and produced viable offspring [48], and surrogate production of Chinese rare minnow (*Gobiocypris rarus*) genome-edited SSC was obtained [38]. The establishment of *C. nasus* SSCs helped to explore SSC gene editing and provides a donor for subsequent studies on germ cell transplantation.

## 5. Conclusions

In this study, we successfully established the SSC line from the testis of the endangered species *C. nasus*, which was induced to differentiate into sperm-like cells through co-culture with *C. nasus* gonadal somatic cells. We observed that the added trophic factors were very effective in maintaining the stable proliferation of spermatogonial stem cells and expanding the culture by borrowing the method of using no feeder layer. This culture method and culture condition can be a model for spermatogonial stem cell cultures in other fish species. The establishment of this spermatogonial stem cell line from the endangered species provides a new donor for studying spermatogenesis in vitro and a foundation for further research on *C. nasus* reproduction and breeding.

## Figures and Tables

**Figure 1 biology-12-01175-f001:**
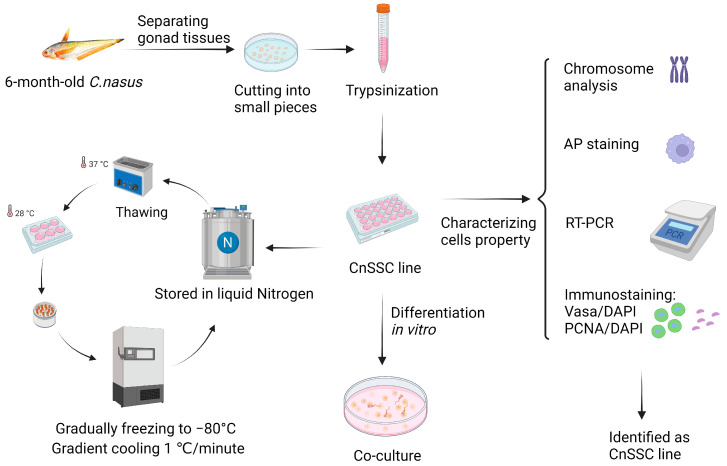
The flow chart of experiments. A total of five 6-month-old *C. nasus* measuring 12 cm in length were accessed. Live fish were put on ice and did not respond to mechanical stimulation. All gonadal tissues were separated, then twice washed in PBS with 1% antibiotics. They were minced and dissociated by trypsinization. The cells were transferred to liquid nitrogen for cryopreservation through appropriate culture. The cells were verified through chromosomal analysis, alkaline phosphatase (AP) staining, RT-PCR, and immunofluorescence and characterized through induction in vitro (the figure was created by Biorcndcr.com).

**Figure 2 biology-12-01175-f002:**
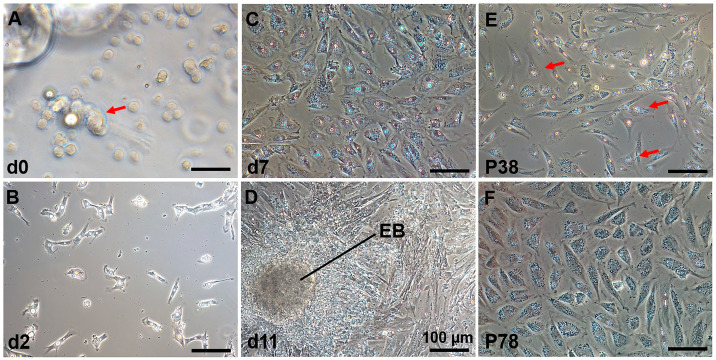
*C. nasus* spermatogonial stem cells (CnSSCs) were cultured in vitro. (**A**) The cells were initially dissociated from the gonadal tissue with a large amount of fat (arrow); (**B**) a small number of *C. nasus* primary testis cells adhered to the plate on the second day; (**C**) the cells adhered to the plates, proliferated, and achieved 80% confluency within seven days; (**D**) primary culture of testis cells on the 11th day. EB: embryoid bodies. (**E**) Cell morphology on the first day of picking spheroid colonies in culture showing a small size and oval or polygonal shape after dissociation (arrow); (**F**) subculture of CnSSC cells at passage 78. (Bars = 20 µm unless indicated.)

**Figure 3 biology-12-01175-f003:**
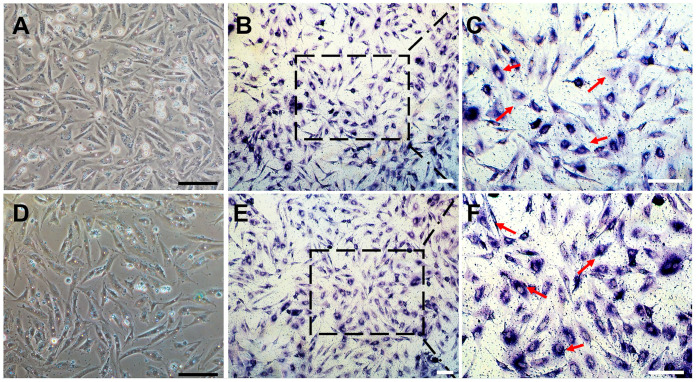
Alkaline phosphatase staining of CnSSCs. (**A**–**C**) Morphology and positive alkaline phosphatase activity staining of cells at P38 (arrow). (**D**–**F**) morphology and positive alkaline phosphatase staining of P38 cells after thawing (arrow). The squares of (**B**(**E**)) are showed in (**C**(**F**)). (Bars = 20 µm.)

**Figure 4 biology-12-01175-f004:**
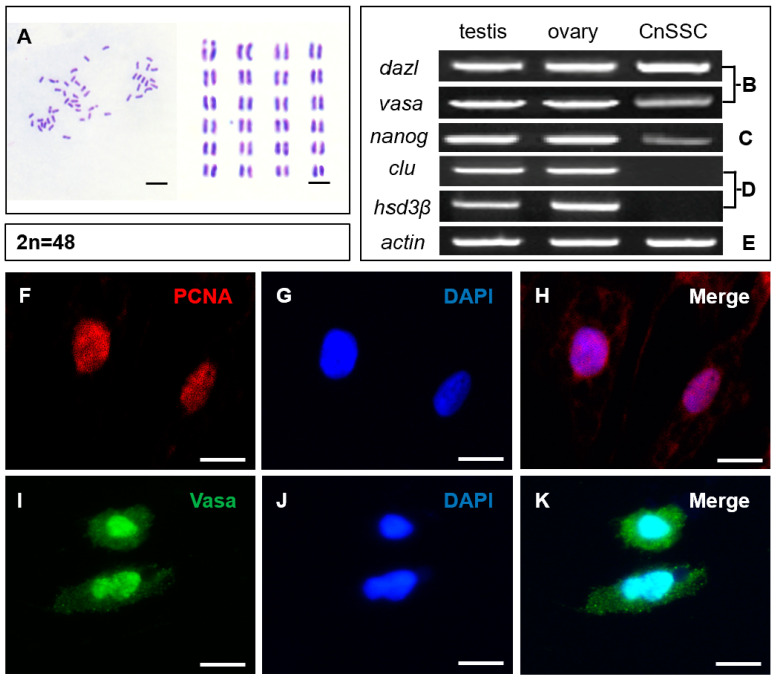
Chromosome analysis, RT-PCR analysis, and immunofluorescence analysis of CnSSCs. (**A**) Diploid metaphase of CnSSCs. (**B**) Expression of germ cell markers, using RT-PCR, of total RNA from CnSSCs and adult tissues with primers for *dazl* and *vasa*. (**C**) Expression of a stem cell marker, using RT-PCR, with primers for *nanog*. (**D**) Expression of somatic cell markers, using RT-PCR, with primers for *clu* and *hsd3β*. (**E**) Expression of actin was determined for calibration. (**F**–**K**) Immunofluorescence of PCNA (red) and Vasa (green) in CnSSCs. (Bars = 10 µm.)

**Figure 5 biology-12-01175-f005:**
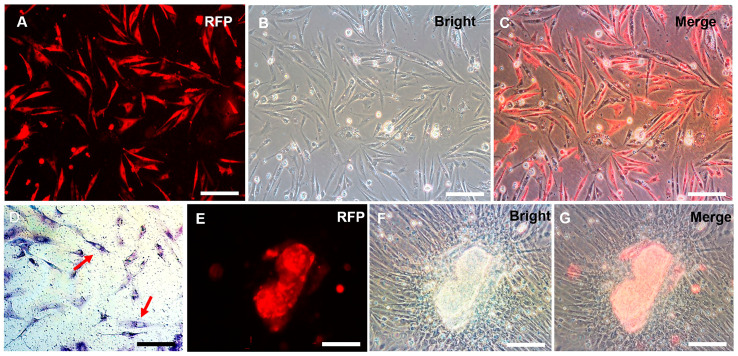
The red fluorescent protein (RFP)-expressing CnSSCs were cultured in vitro. (**A**–**D**) After stable proliferation, CnSSCs cells labeled with RFP were extracted using methods of single-cell clone culturing, exhibiting positive alkaline phosphatase staining (arrow). (**E**–**G**) RFP-expressing CnSSCs formed embryoid bodies by coculture with CnGSCs. (Bars = 20 µm.)

**Figure 6 biology-12-01175-f006:**
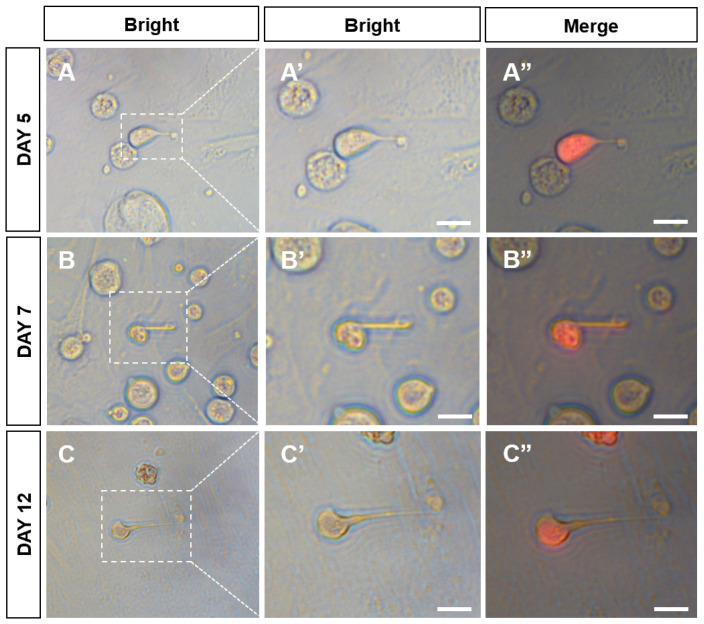
Spermiogenic progression and sperm production from CnSSCs induced in vitro. (**A**–**C**) CnSSC cells differentiated into spherical sperm for 5 days (**A**); the tail pulled longer on the 7th and 12th days (**B**,**C**). (**A’’**–**C’’**) Merge of bright and fluorescent fields. (Bars = 5 µm.)

**Figure 7 biology-12-01175-f007:**
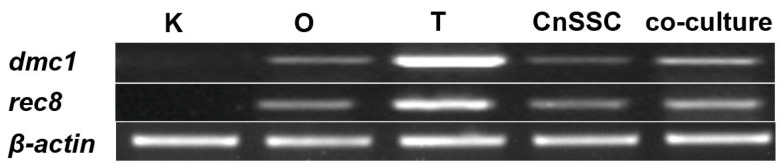
RT-PCR analysis of CnSSCs. Expression of meiotic gene markers, using RT-PCR, of total RNA from CnSSCs and adult tissues with primers for *dmc1* and *rec8* in vitro. Expression of *dmc1* and *rec8* transcripts is low in undifferentiated CnSSCs. CnSSCs express both genes in the coculture condition. K: kidney; O: ovary; T: testis.

**Table 1 biology-12-01175-t001:** Reverse transcription polymerase chain reaction (RT-PCR) primers.

Gene	Primer Sequence	
Name	Forward Primer	Reverse Primer
*dazl*	ACCTGAGGGCAAAATGACACC	CGTGAGCTCCTCTCTTTCATGATGG
*vasa*	ACGCCATCTTCAATCAGTTCCAGACC	CTATTCCCATTCGTCGTCATCTCCGC
*nanog*	ATGGCGGACTGGAAAGTACCAGTAAG	CACAATCTGCAATGCACACAAACATTCAG
*clu*	TCTCTGCTCTGTGTCTTATC	AACTTCTTGTGGTCCTCTC
*hsd3β*	GTGGTGGTGGTAGCGAAGT	GCCTCCGACAGCATACAGT
*dmc1*	TGTCACCAACCAGATGACGG	TTGGCATCCGTGATTCCTCC
*rec8*	CCGAGTCTGCCTAAACCACG	CTTTCTCCTTAAGAGTGATG
*β-actin*	TTCAACAGCCCTGCCATGTAC	CCTCCAATCCAGACAGAGTATT

## Data Availability

Not Applicable.

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
