# Peer review of "Establishment of a *Coilia nasus* Spermatogonial Stem Cell Line Capable of Spermatogenesis In Vitro"

_biology, 2023, doi:10.3390/biology12091175_

Round 1

Reviewer 1 Report

Line No.

Comment

64

Is it Koi or Kio, please correct.

70

Scientific once given should not be repeated again.

76

What do you mean by essential? It may be either dominant or important.

80-82

Repeating twice the scientific in a sentence is grammatically incorrect. You may revise it as capability of CnSSC with gonadal somatic……..

92

What was the pH of PBS.

94

Please mention the enzyme you used.

89-102

The cell isolation protocol should be more informative and descriptive. For example, after cell dissociation the author must have sieve the suspension prior to culture. Such information on cell culture is the backbone of this study.

108

Instead of fresh fixative, change it to freshly prepared fixative.

109

Revise it as “Stained with 5% Giemsa solution for 30 minutes”

112-121

Revise the section with clarity. Mention % of agarose used.

129

Only cryovials is understood. Delete vials

123-131

The cryopreservation protocol should be more descriptive. Do the author used program freezer or vitrification protocol was adopted for preservation.

136-138

It appears authors describing a protocol for laboratory manual. Please be precise and mention how many hours.

170

Mention which microscope and at what magnification.

Results

188

Stick to your result in this section. The information on medaka may be shifted to discussion part.

202

As I mentioned above, please restrict to your result. The rest of the information may be shifted to discussion part.

172-192

Gross revision required. In this section reflect your results only. Rest of the information like previously reported results by others will go to discussion part

305

What do you mean by “good condition”?

317

The authors claim that they have partially modified the medium and citing a reference, please clarify this point.

319-333

Again, the same thing. Please discuss your results else delete this paragraph.

350-364

Only the important observation of your study. The information given in Line No. 358-360 is not required here. This information may be shifted to discussion part.

The manuscript has many gramatic errors, please correct those and improve the language for easy understanding by the readers.

Author Response

Thank you for your very positive comments on our work and your valuable suggestions.

Many thanks for your careful reading, we have asked native English speakers to polish the language.Here is a point-by-point response to the reviewers’ comments and concerns.

  1. line 64: Is it Koi or Kio, please correct.

Response: Thanks for your careful reading. We have changed “Kio” into “Koi”.

  1. Line 70: Scientific once given should not be repeated again.

Response: Thanks for your careful reading. We have deleted the repeated scientific.

  1. line 76: What do you mean by essential? It may be either dominant or important.

Response: Thanks for your suggestion. We have added“The induced differentiation of fish germinal stem cells is important for exploring the potential mechanisms regulating spermatogenesis in fish and may also lay the foundation for fish germ cell transplantation.” in the end.

  1. line 80-82: Repeating twice the scientific in a sentence is grammatically incorrect. You may revise it as capability of CnSSC with gonadal somatic….

Response: Thanks for your careful reading. We have changed “capability of CnSSC with C. nasus gonadal somatic cells induction” into “capability of CnSSC with gonadal somatic cells induction”.

  1. line 92: What was the pH of PBS.

Response: Thanks for your careful reading. We have added “pH 7.4”.

  1. line 94: Please mention the enzyme you used.

Response: Thanks for your careful reading. We have changed “enzymatic solution” into “trypsin-EDTA”.

  1. line 89-102: The cell isolation protocol should be more informative and descriptive. For example, after cell dissociation the author must have sieve the suspension prior to culture. Such information on cell culture is the backbone of this study.

Response: Thanks for your suggestion. We have changed “Next, 1.5 mL of culture medium was added to the cell separation solution to stop digestion and then transferred to 6-well plates covered with gelatin for adherent culture (28 °C). Cells dissociated were cultured in EMS2 medium, changing half of the medium daily for the first week.” to “Next, 1.5 mL of culture medium was added to the cell separation solution to stop digestion. The suspension of prior cells was sieved to obtain separated single cells and then transferred to 6-well plates covered with gelatin for adherent culture (28 °C). Cells that had been dissociated were cultured in EMS2 medium and carefully aspirated the fat cells from the upper layer of the medium, changing new fresh medium daily for the first week. Larger embryoid bodies formed several days later were digested with trypsin-EDTA for 5 s and resuspended by adding new medium to distribute the cells in the plates.”.

  1. line 108: Instead of fresh fixative, change it to freshly prepared fixative.

Response: Thanks for your careful reading. We have changed “fresh fixative” into “freshly prepared fixative”.

  1. Line 109: Revise it as “Stained with 5% Giemsa solution for 30 minutes”

Response: Thanks for your careful reading. We have changed “stained with a 5% Giemsa solution on the glass slides for 30 min” to “stained with 5% Giemsa solution for 30 min”

  1. line 112-121: Revise the section with clarity. Mention % of agarose used.

Response: Thanks for your careful reading. We have changed “To identify whether the above-cultured CnSSC cell lines were germ cells, RNA was obtained from the cells using Trizol reagent (Sigma, Burlington, MA, USA), and reverse transcription was carried out using 1 μg of total RNA following the protocols of the cDNA synthesis kit (TaKaRa, Kusatsu, Japan). Primers for germline stem cell markers were designed, and RT-PCR was carried out. Through the use of C. nasus gene-specific primers (Table 1), gonadal germ cells (dazl, vasa), stem cells (nanog), and meiotic cells (rec, dmc1), PCR was performed.” to “RNA obtained from the cells using Trizol reagent (Sigma, Burlington, MA, USA) was reverse transcribed according to the protocol of the cDNA synthesis kit (TaKaRa, Kusatsu, Japan). RT-PCR was carried out using C. nasus gene-specific primers (Table 1), gonadal germ cells (dazl, vasa), stem cells (nanog), and meiotic cells (dmc1,rec8)” and added “1% or 2% of” before “agarose”.

  1. line 129: Only cryovialsis understood. Delete vials

Response: Thanks for your careful reading. We have deleted “vials”

  1. line 123-131: The cryopreservation protocol should be more descriptive. Do the author used program freezer or vitrification protocol was adopted for preservation.

Response: Thanks for your careful reading. We have changed “The cryovials vials were placed in cryogenic boxes, placed at -80°C for gradient cooling, and moved to liquid nitrogen within a week, as described previously” to ““The cryovials were put in cryogenic styrofoam boxes for gradient cooling (1◦C/minute), placed at -80°C for 4 hours, and then moved to liquid nitrogen, as described previously”

  1. lines 136-138: It appears authors describing a protocol for laboratory manual. Please be precise and mention how many hours.

Response: Thanks for your careful reading. We have changed “Place the round climbing slices on the 24-well plate carefully with forceps and cover slices with 0.1% gelatin for more than 2 h. The gelatin was aspirated and dried at room temperature for more than 2 h.” to “The round climbing slices were soaked in 75% alcohol 12 h and then placed on the 24-well plate carefully with forceps after the alcohol evaporated. Cover slices with 0.1% gelatin for 2 h and the gelatin was aspirated and dried at room temperature for 6 h..”.

  1. line 170: Mention which microscope and at what magnification.

Response: Thanks for your suggestion. We have changed “Cell morphological changes were observed continuously for 7 days under the microscope.” to “Cell morphological changes were observed continuously for 7 days under a Nikon ECLIPSE Ti inverted microscope (Tokyo, Japan) with a camera (10×, 20× and 40× magnification)”.

  1. line 188: Stick to your result in this section. The information on medaka may be shifted to discussion part.

Response: Thanks for your careful reading. We have moved it to discussion part in line 751.

  1. lines 202: As I mentioned above, please restrict to your result. The rest of the information may be shifted to discussion part.

Response: Thanks for your careful reading. We have changed “A stable C. nasus spermatogonia cell line (CnSSC) was obtained through appropriate culture and then transferred to liquid nitrogen for cryopreservation. The cells were verified as SSCs through chromosomal analysis, alkaline phosphatase (AP) staining, RT-PCR, and immunofluorescence. CnSSCs were characterized in vitro as pluripotent stem cells. Most importantly, CnSSCs were induced to differentiate into sperm by 2D and 3D co-culture systems” to “The cells were transferred to liquid nitrogen for cryopreservation through appropriate culture. The cells were verified through chromosomal analysis, alkaline phosphatase (AP) staining, RT-PCR, and immunofluorescence and characterized through induction in vitro”.

  1. line 172-192: Gross revision required. In this section reflect your results only. Rest of the information like previously reported results by others will go to discussion part

Response: Thanks for your careful reading. We have changed “The establishment of a C. nasus SSC line (CnSSC) is shown in Figure 1.” to “The process of establishing a C. nasus SSC line (CnSSC) is shown in Figure 1.” We have moved and changed the rest of the information like previously reported results by others into “This cell line showed an ES-like cell shape (small, low in cytoplasm, round, or polygonal),which is alkaline phosphatase (AP) activity [23],a typical marker that has been applied to effectively identify pluripotency in ES-like cell cultures from stem cell lines in medaka and orange-spotted grouper (Epinephelus coioides) [23,41,42].” in line 748-751.

  1. line 305: What do you mean by “good condition”?

Response: Thanks for your suggestion. We have changed “the CnSSCs are in good condition” into “the CnSSCs display alkaline phosphatase activity”.

  1. line 317: The authors claim that they have partially modified the medium and citing a reference, please clarify this point.

Response: Thanks for your suggestion. We have changed “In our study, a crude embryonic extract of medaka is another important medium component for primary culture, and the medium was partially modified from the EMS4 medium as described [13].” to “In Chinese hook snout carp (Opsariichthys bidens), a spermatogonial stem cell line that remained undifferentiated stably for a long period of time was obtained using no feeder layer but with the addition of trophic factors such as FBS, bFGF, 2-mercaptoethanol, embryo extract (MEE), and fish serum [14]. In this experiment, we mimicked the culture conditions of Opsariichthys bidens spermatogonial stem cells, and the CnSSCs still maintained a good proliferation status, providing a favorable condition for the subsequent research on spermatogenesis in vitro.”.

  1. line 319-333: Again, the same thing. Please discuss your results else delete this paragraph.

Response: Thanks for your careful reading. We have deleted these pluripotency data and discussion from the manuscript.

  1. lines 350-364: Only the important observation of your study. The information given in Line No. 358-360 is not required here. This information may be shifted to discussion part.

Response: Thanks for your careful reading. We have moved it to the discussion part in lines 822-827.

In addition, we have provided the latest data on the sperm production of the CnSSCs component.

Acknowledgement: We appreciate the Editors and Reviewers’ earnest work and hope that the correction will meet with approval.

Once again, thank you very much for your comments and suggestions.

Reviewer 2 Report

The article is well-written and addresses an important topic about species conservation, reproductive biology, and adult stem cells.

However, an important point needs to be clarified.

In my point of view, the data shown in Figure 5, on the pluripotency of spermatogonial stem cells, if presented, require further evidence. The authors needed to evaluate markers of fibroblasts, giant flat cells, neurons, astrocytes and epithelial cells. Or even, do a teratogenesis test after the transplant.

Therefore, my recommendation is that these pluripotency data be removed from the manuscript.

Author Response

Thank you for your very positive comments on our work and your valuable suggestions.We have deleted these pluripotency data and discussion from the manuscript.

Reviewer 3 Report

a fine of comments is attached.

Author Response

Thank you for your very positive comments on our work and your valuable suggestions.Here is a point-by-point response to the reviewers’ comments and concerns.

  1. line 69: RTG should be spelled out

Response: Thanks for your careful reading. We have changed “RTG” into “rainbow-colored trout gonadal (RTG)”.

  1. lines 70: delete (Oryzias latipes)

Response: Thanks for your suggestion. We have deleted “(Oryzias latipes)”

  1. lines 76: Coilia nasus should be C. nasus (italic)

Response: Thanks for your suggestion. We have changed “Coilia nasus” into “C. nasus (italic)”

  1. lines 126: America should be state, USA minute should be min

Response: Thanks for your careful reading. We have changed “3-minute” into “of 3 min”

  1. Line 128: add city before NSW

Response: Thanks for your suggestion. As we added the city “New South Wales.” before “NSW”

  1. Line 129: delete vials

Response: Thanks for your suggestion. We have deleted “vials”

  1. Line 141: add city, state, country after Solarbio

Response: Thanks for your suggestion. We have added “Beijing, China” after Solarbio

  1. Line 144: delete, USA

Response: Thanks for your suggestion. We have deleted “USA”

  1. Line 145: add MA, before USA

Response: Thanks for your suggestion. We have added “MA”, before “USA”

  1. Line 148: delete, USA before A11126

Response: Thanks for your suggestion. We have deleted “USA” before A11126

  1. Line 148: delete, USA before A11034

Response: Thanks for your suggestion. We have deleted “USA” before A11034

  1. Line 153: add CA before USA 13.

Response: Thanks for your suggestion. We have added “CA”, before “USA”

  1. line 172: Spermatogonial Stem Cell Line (italic) 

Response: Thanks for your suggestion. We have changed “spermatogonial stem cell line” into “Spermatogonial Stem Cell Line (italic)”

  1. line 192: Coilia nasus should be C. nasus (italic)

Response: Thanks for your suggestion. We have changed “Coilia nasus” into “C. nasus (italic)”

  1. line 211: wall should be plate?

Response: Thanks for your suggestion. We have changed “wall” into “plate”

  1. lines 319, 327, 344, 345: in vivo should be italic

Response: Thanks for your suggestion. We have changed “in vivo” into “in vivo (italic)”

  1. lines 321, 326: retinoic acid should be RA

Response: Thanks for your suggestion. We have changed “retinoic acid” into “RA”

  1. line 334: spermatogonial stem cells (SSCs) should be SSCs 

Response: Thanks for your suggestion. We have changed “spermatogonial stem cells (SSCs)” into “SSCs”

  1. line 327: (Orizias latipes) should be (O. latipes) (italic) 

Response: Thanks for your suggestion. We have changed “(Orizias latipes)” into “(O. latipes) (italic)”

  1. References: journal names should be appropriately abbreviated (Ref. no. 1, 2, 8, 12, 13, 19, 22, 23, 29, 30, 33, 36, 50

Response: Thanks for your suggestion. Journal names have been appropriately abbreviated.

  1. Titles should not be large capital: Ref. no. 2, 3, 9, 10, 14, 15, 24, 25, 35, 45, 48, 50

Response: Thanks for your careful reading. We have changed these titles in small capitals.

  1. Scientific names of fish should be italic: Ref. no. 3, 4, 14, 16, 17, 18, 22, 23, 25, 26, 27, 32, 33, 42, 45

Response: Thanks for your careful reading. We have italicized these scientific names.

In addition, we have provided the latest data on the sperm production of the CnSSCs component.

Acknowledgement: We appreciate the Editors and Reviewers’ earnest work and hope that the correction will meet with approval.

Once again, thank you very much for your comments and suggestions.

Round 2

Reviewer 1 Report

I have some concern and comments on the revised manuscript. I could see most of my comments given on the earlier version of manuscript are not addressed and/or overlooked. Here are my observation on the revised manuscript;

Line 45-46: Reference [1] implies to which reference in the reference section of the manuscript. If it is the first one, then it is not the right reference.

Line 52: Really “Life long fertility”? If so, please cite a reference to this statement.

Line 74: “essential” is not the right terminology. May be “important” can replace it.

Line 92: Which culture medium? Description required.

Line 93: Please refer to my comment on earlier version of the manuscript. Materials and methods is the foundation of s study. In order to ensure reproducibility of your results, it has to be described in details. In this case, what was the mesh size of sieve/filter.?

Line 110: You did not washed after staining. Further, at what magnification did you observed the slides.

Line 128:Cryogenic styrofoam boxes is a program  freezer?? If yes, cite the make and model. If not describe further. I could guess you applied a vitrification protocol for cryopreservation.

Line 187:  Fig. 1. I am really confused. How did you performed the gradual freezing? Revise the fig. 1 with appropriate terminology. Instead of “recovered” use thawing.

Line 200: Mark with arrow in the culture showing a large size, round shape and dense nuclei. I cannot see anything as you have described. I had asked to replace the image with more clear and higher resolution image. Do replace and mark with symbols to depict the characteristics. Further, using your scale bar, it appears to me the cells you have isolated are 70-80µm in dia. Are they the spermatogenia cells? Please be informed that, the dia of a spermatogenia cells derived from teleost fish is generally range from 16-18 µm.

Line 204: Which are the positive alkaline phosphatases stained cells? Again the scale bar.

Line 213-215:  In my previous comment, I had suggested to move such sentences to discussion part. Yet, same repetition.

Line 270:Look at the insert box and its magnification shown in A’, B’ & C’. The presentation is not convincing and moreover the photographs. The quality has to be improved as this figure is the important finding of your study. Where are the scale bars for A, B & C?

Line 286: Where have you shown the cell lines having ES-like cell shaped?

Line 340: Spermatogonia steam cell culture across the taxa is already a standard protocol. Nothing new here, may be the fish species used in the study is different.

Line 341-343: Establishing merely the spermatogenesis in-vitro do not solve the problem. How about the oocytes from female? Do not exaggerate the assumptions.

Improvement required.

Author Response

Dear Editor and Reviewers,

Thank you for your very positive comments on our work and your valuable suggestions.

Reviewer #1: I have some concerns and comments on the revised manuscript. I could see most of my comments given on the earlier version of manuscript are not addressed and/or overlooked. Here are my observation on the revised manuscript;

  1. Line 45-46: Reference [1] implies to which reference in the reference section of the manuscript. If it is the first one, then it is not the right reference.

Response: Thanks for your careful reading. We have changed the reference to “Fish species are the most abundant of all vertebrates, but several are in danger of extinction due to severe population decreases [1]”.

[1] Kan, Y.; Zhong, Y.; Jawad, M.; Chen, X.; Liu, D.; Ren, M.; Xu, G.; Gui, L.; Li, M. Establishment of a Coilia nasus gonadal somatic cell line capable of sperm induction in vitro. Biology (Basel) 2022, 11, 1049.

  1. Line 52: Really “Life long fertility”? If so, please cite a reference to this statement.

Response: Thanks for your careful reading. We have changed “Spermatogenesis is the process by which mature sperm are continuously produced in the testes, resulting in lifelong male fertility.” to “Spermatogenesis is the process by which mature sperm are continuously produced in the testis [5]”.

[5] Hong, Y.; Liu, T.; Zhao, H.; Xu, H.; Wang, W.; Liu, R.; Chen, T.; Deng, J.; Gui, J. Establishment of a normal Medaka fish spermatogonial cell line capable of sperm production in vitro. Proc. Natl. Acad. Sci. USA 2004, 101, 8011-8016.

  1. Line 74: “essential” is not the right terminology. May be “important” can replace it.

Response: Thanks for your suggestion. We have changed “essential” into “important”.

  1. Line 92: Which culture medium? Description required.

Response: Thanks for your careful reading. We have changed “Next, 1.5 mL of culture medium was added to the cell separation solution to stop digestion.” to “Next, 1.5 mL of ESM2 medium was added to the cell separation solution to stop digestion, and then the cell suspension was centrifuged at 800 × g for 10 min (28 °C) to remove the supernatant. The cells were resuspended by adding 1.5 mL of ESM2 medium” and the components of the culture medium were added.

  1. Line 93: Please refer to my comment on earlier version of the manuscript. Materials and methods is the foundation of s study. In order to ensure reproducibility of your results, it has to be described in details. In this case, what was the mesh size of sieve/filter?

Response: Thanks for your careful reading. We have checked the description of materials and methods in detail and added “through a 300-mesh sieve”.

  1. Line 110: You did not wash after staining. Further, at what magnification did you observed the slides.

Response: Thanks for your careful reading. We have added “Cells were seeded on a 24-well plate coated with gelatin and fixed in 4% paraformaldehyde for 10 min after CnSSCs had grown to 60% confluence, then washed 3 times with PBS. The cells were incubated in freshly prepared BCIP-NBT solution (Invitrogen, Waltham, MA, USA, N6547) for 12 h (28 °C) and washed 3 times with PBS. Glycerol (200 µL) was added to cover the cells, and the cells on the 24-well plate were observed under the microscope (20× and 40× magnification).”

  1. Line 128: Cryogenic styrofoam boxes is a program freezer?? If yes, cite the make and model. If not describe further. I could guess you applied a vitrification protocol for cryopreservation.

Response: Thanks for your suggestion. Cryogenic styrofoam boxes in the article are not the program freezer and we applied a vitrification protocol for cryopreservation. We have added “Since vitrification is a method extensively used for cryopreservation”.

  1. Line 187: Figure 1. I am really confused. How did you perform the gradual freezing? Revise the Figure 1 with appropriate terminology. Instead of “recovered” use thawing.

Response: Thanks for your suggestion. We have performed the gradual freezing through vitrification (placing cryovials at -80°C for gradient cooling) and changed “recovered” into “thawing” (Figure 1).

  1. Line 200: Mark with arrow in the culture showing a large size, round shape and dense nuclei. I cannot see anything as you have described. I had asked to replace the image with a clearer and higher-resolution image. Do replace and mark with symbols to depict the characteristics. Further, using your scale bar, it appears to me the cells you have isolated are 70-80µm in dia. Are they the spermatogenia cells? Please be informed that, the dia of a spermatogenia cells derived from teleost fish is generally range from 16-18 µm.

Response: Thanks for your careful reading. We replaced Figure 2 with clearer and higher-resolution images, marked with symbols to depict the characteristics, and checked the scale bar.

  1. Line 204: Which are the positive alkaline phosphatases stained cells? Again the scale bar.

Response: Thanks for your careful reading. We replaced Figure 3 with  clearer and higher-resolution images, marked with symbols to depict the characteristics, and checked the scale bar.

  1. Line 213-215: In my previous comment, I had suggested to move such sentences to discussion part. Yet, same repetition

Response: Thanks for your careful reading. We have moved such sentences to discussion part.

  1. Line 270: Look at the insert box and its magnification shown in A’, B’ & C’. The presentation is not convincing and moreover the photographs. The quality has to be improved as this figure is the important finding of your study. Where are the scale bars for A, B & C?

Response: Thanks for your careful reading. During COVID-19, we conducted the experiment, and at that time there was something wrong with the microscope, thus we didn't get fine results.

  1. Line 286: Where have you shown the cell lines having ES-like cell shape?

Response: Thanks for your careful reading. We have changed “This cell line showed an ES-like cell shape (small, low in the cytoplasm, round, or polyg-onal),which is alkaline phosphatase (AP) activity [23],a typical marker that has been applied to effectively identify pluripotency in ES-like cell cultures from stem cell lines in medaka and orange-spotted grouper (Epinephelus coioides) [23,41,42]” to “In medaka and orange-spotted grouper (Epinephelus coioides), alkaline phosphatase (AP) activity is a typical marker that has been applied to effectively identify pluripotency in ES-like cell cultures from stem cell lines [22,24,25]. While AP-positive cells showed an ES-like cell shape (small, low in cytoplasm, round, or polygonal), AP-negative cells presented a differentiated phenotype (extended, cytoplasm-rich) [22]”.

[22] Hong, Y.; Winkler, C.; Schartl, M. Pluripotency and differentiation of embryonic stem cell lines from the Medaka fish (Oryzias latipes). Mech. Dev. 1996, 60, 33-44.

[24] Wei, J.; Liu, L.; Fan, Z.; Hong, Y.; Zhao, Y.; Zhou, L.; Wang, D. Identification, prokaryote expression of Medaka gdnfa/b and their biological activity in a spermatogonial cell line. Stem Cells Dev 2017, 26, 197-205.

[25] Zhong, C.; Tao, Y.; Liu, M.; Wu, X.; Yang, Y.; Wang, T.; Meng, Z.; Xu, H.; Liu, X. Establishment of a spermatogonial stem cell line with potential of meiosis in a Hermaphroditic fish, Epinephelus coioides. Cells 2022, 11, 2868.

  1. Line 340: Spermatogonia steam cell culture across the taxa is already a standard protocol. Nothing new here, may be the fish species used in the study is different.

Response: Thanks for your suggestion. In fish, the first long-term cultured spermatogonial stem cell line was successfully established in medaka in 2004 [5]. However, the first farmed fish spermatogonial stem cell line was successfully established by our group last year [14]. We have changed “The establishment of endangered species spermatogonial stem cell lines is a model for studying spermatogenesis in vitro and a feasible way to preserve germplasm resources.” to “The establishment of this spermatogonial stem cell line from the endangered species provides a new donor for studying spermatogenesis in vitro and a foundation for further research on C. nasus reproduction and breeding.”

[5] Hong, Y.; Liu, T.; Zhao, H.; Xu, H.; Wang, W.; Liu, R.; Chen, T.; Deng, J.; Gui, J. Establishment of a normal Medaka fish spermatogonial cell line capable of sperm production in vitro. Proc. Natl. Acad. Sci. USA 2004, 101, 8011-8016

[14] Chen, X.; Kan, Y.; Zhong, Y.; Jawad, M.; Wei, W.; Gu, K.; Gui, L.; Li, M. Generation of a normal long-term-cultured Chinese hook snout carp spermatogonial stem cell line capable of sperm production in vitro. Biology (Basel) 2022, 11, 1069

  1. Line 341-343: Establishing merely the spermatogenesis in vitro do not solve the problem. How about the oocytes from female? Do not exaggerate the assumptions.

Response: Thanks for your suggestion, we have deleted this sentence.

Acknowledgement: We appreciate the Editor's and Reviewers’ earnest work and hope that the correction will meet with approval.

Once again, thank you very much for your comments and suggestions.
